# Spectroradiometer Calibration for Radiance Transfer Measurements

**DOI:** 10.3390/s23042339

**Published:** 2023-02-20

**Authors:** Clemens Rammeloo, Andreas Baumgartner

**Affiliations:** German Aerospace Center (DLR), Remote Sensing Technology Institute (IMF), 82234 Weßling, Germany

**Keywords:** spectrometer, radiometric calibration, spectral calibration, temperature sensitivity, polarization sensitivity, radiance transfer, SVC HR-1024i

## Abstract

Optical remote sensing and Earth observation instruments rely on precise radiometric calibrations which are generally provided by the broadband emission from large-aperture integrating spheres. The link between the integrating sphere radiance and an SI-traceable radiance standard is made by spectroradiometer measurements. In this work, the calibration efforts of a Spectra Vista Corporation (SVC) HR-1024i spectroradiometer are presented to study how these enable radiance transfer measurements at the Calibration Home Base (CHB) for imaging spectrometers at the Remote Sensing Technology Institute (IMF) of the German Aerospace Center (DLR). The spectral and radiometric response calibrations of an SVC HR-1024i spectroradiometer are reported, as well as the measurements of non-linearity and its sensitivity to temperature changes and polarized light. This achieves radiance transfer measurements with the calibrated spectroradiometer with relative expanded uncertainties between 1% and 3% (k=2) over the wavelength range of 380 nm to 2500 nm, which are limited by the uncertainties of the applied radiance standard.

## 1. Introduction

Spectroradiometers are ubiquitous instruments in optical calibration laboratories as well as optical remote sensing and Earth observation. These instruments are widely used for measuring spectral irradiance and reflectance [1] and require characterization and calibration efforts to enable precise measurements [2,3,4]. In this paper, we report on the calibration of an HR-1024i spectroradiometer from the Spectra Vista Corporation (SVC) and its application in radiance transfer measurements. This spectroradiometer model provides reflectance spectra for validating airborne and satellite hyperspectral data [5,6] and the non-destructive measurements of plant traits [7,8,9,10]. On the other hand, an HR-1024i spectroradiometer is applied as a radiance transfer spectroradiometer in the Calibration Home Base (CHB) of the Remote Sensing Technology Institute (IMF) at the German Aerospace Center (DLR) [11] for the radiometric calibration of several hyperspectral imaging systems such as the Airborne Prism Experiment (APEX) [12], the Munich Aerosol Cloud Scanner (specMACS) [13], and the Environmental Mapping and Analysis Program (EnMAP) [14].

In all of these calibration measurements, an HR-1024i spectroradiometer with a custom fore-optics is first calibrated on an SI-traceable radiance standard. This spectroradiometer then measures the spectral radiance of a large-aperture integrating sphere that is illuminated by multiple quartz tungsten halogen (QTH) lamps. Besides a radiometric and spectral calibration of the spectroradiometer response, other instrumental effects also need to be characterized to reach the required uncertainty of the radiance measurements in the range of 1–2%.

In this work, we first describe the spectroradiometer and its calibration setup in Section 2. Next, a model of the instrument response is presented in Section 3, followed by its spectral response function as determined from spectral calibration measurements in Section 4. Section 5, Section 6, Section 7 and Section 8 discuss the measured instrument parameters that are needed for the instrument model, and the related uncertainty contributions which could impact both laboratory and field measurements. Finally, an example application of radiance transfer measurements on a large-aperture integrating sphere is presented in Section 9.

## 2. Spectroradiometer Setup

The HR-1024i spectroradiometer has three detectors with a total of 1024 spectral channels, that are specified to cover wavelengths from 350 nm to 2500 nm [15]. The instrument used in this work has a slightly larger wavelength range, as given in Table 1.

A lens fore-optic is commonly used in field measurements, which gives the spectroradiometer a 4 ${}^{{}^{\circ}}$ field-of-view (FOV). In the laboratory of the CHB, however, a 1 m fiber optic bundle is combined with an off-axis parabolic (OAP) mirror (Thorlabs MPD249-F01) to aid the precision alignment of the spectroradiometer FOV. This alignment is performed on the DLR radiance standard (RASTA) for radiometric calibrations and sketched in Figure 1. Instead of connecting the fiber optic bundle to the spectroradiometer, light from a diffuse LED source is coupled into the fiber bundle. This creates a beam with an $1/e^{2}$-diameter of 5 cm after the OAP mirror that represents the spectroradiometer’s FOV. By adjusting the mirror mount, the FOV can thus be aligned to the area of interest.

RASTA consists of a Spectralon reflectance panel at a fixed position from an FEL lamp. The spectral radiance reflected from the panel at a 45${}^{{}^{\circ}}$ angle from the optical axis is calibrated to SI-traceable standards [16]. The stability of the radiance standard is monitored during the measurements by a set of five filter radiometers that each cover a complementary part of the calibration wavelength range from 300 nm to 2500 nm. A detailed description of RASTA is found in [17]. For the radiometric responsivity calibration, the spectroradiometer can be placed inside a temperature-controlled enclosure, such that the calibrations can be performed under different environmental temperatures. The results of this radiometric calibration are described in Section 7.

## 3. Instrument Model

The response of each spectral channel *c* of the spectroradiometer is modeled here for an homogeneous irradiance of the input optics. The measurement equation of the spectroradiometer signal $s_{c}$ in digital units DN to an at-aperture spectral radiance $L_{\lambda }\left(\lambda \right)$ is
(1)$$s_{c}=R_{c}t_{\mathrm{int},d}\left[1+C_{T,c}\left(T_{d}-T_{\mathrm{ref}}\right)\right]\;\int_{0}^{\infty }\;\;G_{c}\left(\lambda -\lambda_{c}\right)L_{\lambda }\left(\lambda \right)d\lambda +s_{\mathrm{dark},c}\left(T_{d},t_{\mathrm{int},d}\right).$$

In this equation, $R_{c}$ is the radiometric responsivity of a channel *c* in units of DN mW${}^{-1}$ ms${}^{-1}$ m${}^{2}$ nm sr. The responsivity depends on the fore-optic installed on the spectroradiometer and is determined from calibration measurements with the DLR radiance standard at specific integration times $t_{\mathrm{int},d}$ for each detector array. Because the VNIR detector is not temperature-stabilized, its response changes with the instrument’s operating temperature. This effect is modeled as a linear relation in Equation (Equation 1) with the difference of the detector temperature $T_{d}$ with respect to the reference temperature $T_{\mathrm{ref}}$ of the detector during the radiometric calibration, where $C_{T,c}$ is the temperature sensitivity coefficient. The normalized spectral response function (SRF) $G_{c}\left(\lambda -\lambda_{c}\right)$ with the center wavelength $\lambda_{c}$ for each channel is determined from monochromator measurements as will be discussed in Section 4.

The dark signal $s_{\mathrm{dark},c}$ in Equation (Equation 1) can also be described with a linear function of both the detector temperature $T_{d}$ and integration time $t_{\mathrm{int},d}$. However, the HR-1024i spectroradiometer performs a dark measurement with an internal shutter immediately before or after each light measurement and supplies the operator with the dark subtracted signal $s_{c}-s_{\mathrm{dark},c}$. In practice, repeated measurements are averaged, and therefore we describe the spectroradiometer measurements using the signal mean $S_{c}$:(2)$$S_{c}=\langle s_{c}-s_{\mathrm{dark},c}\rangle =R_{c}t_{\mathrm{int},d}\langle \left[1+C_{T,c}\left(T_{d}-T_{\mathrm{ref}}\right)\right]\;\int_{0}^{\infty }\;\;G_{c}\left(\lambda -\lambda_{c}\right)L_{\lambda }\left(\lambda \right)d\lambda \rangle .$$

The standard uncertainty of the signal mean $u\left(S_{c}\right)$ depends on both the instrument noise and any instability of the spectral radiance. The instrument noise of the dark-signal-subtracted signal is found to be in the range of 2–4 DN (RMS) for observation times of 1 s. The effective signal-to-noise ratio $S_{c}/u\left(S_{c}\right)$ can be increased simply by averaging repeated measurements. We found that optimum measurement durations are in the range of 20–50 s when measuring QTH lamp sources in our laboratory. Longer measurement series are typically not limited by the spectroradiometer noise, but are instead dominated by other factors, such as a drift in the light source radiance.

The parameters $R_{c}$ and $C_{T,c}$ and the SRF $G_{c}\left(\lambda -\lambda_{c}\right)$ in Equation (Equation 2) need to be determined from calibration measurements, which are discussed in the following sections.

## 4. Spectral Response Function

A spectral calibration of the spectroradiometer needs to be performed both for reflectance and for radiance transfer measurements. The spectral response function is measured with a monochromator (Oriel Instruments MS257) with an uncertainty in its central wavelength of 0.1 nm. This is achieved by calibrating the monochromator to the 435.6 nm spectral line of an mercury-vapor lamp following the method described in [18]. The full-width at half-maximum (FWHM) bandwidth of the monochromator ranges from 0.5 to 1.2 nm for the wavelengths of the VNIR detector array and between 1.2 and 2.5 nm for the range covered by the SWIR detector arrays. Since the monochromator bandwidth is between 4 and 10 times smaller than the FWHM bandwidth of the spectroradiometer SRF, the measurement equation is simplified to
(3)$$S_{c}=R_{c}t_{\mathrm{int},d}\langle \left[1+C_{T,c}\left(T_{d}-T_{\mathrm{ref}}\right)\right]L\left(\lambda_{\mathrm{mono}}\right)\rangle G_{c}\left(\lambda_{\mathrm{mono}}-\lambda_{c}\right),$$
where $L(\lambda_{\mathrm{mono}})$ is the monochromator radiance at a central wavelength $\lambda_{\mathrm{mono}}$. The average is assumed to only include the terms that vary during the repeated measurements, that is the VNIR detector temperature $T_{d}$ and the monochromator radiance $L(\lambda_{\mathrm{mono}})$.

Equation (Equation 3) shows that the SRF $G_{c}\left(\lambda \right)$ for each channel is obtained by scanning the monochromator wavelength, as illustrated in Figure 2. The normalized SRF is parameterized with an asymmetric Gaussian function [19]:(4)$$G_{c}\left(\lambda \right)=\left\{\begin{array}{ccc} \frac{1}{\sigma_{c}\sqrt{2\pi }}\exp\;\left[-\frac{1}{2}{\left(\frac{\lambda -\lambda_{c}}{\sigma_{c}-\alpha_{c}}\right)}^{2}\right] & \mathrm{for} & \lambda <\lambda_{c}\\ \; & \; & \\ \frac{1}{\sigma_{c}\sqrt{2\pi }}\exp\;\left[-\frac{1}{2}{\left(\frac{\lambda -\lambda_{c}}{\sigma_{c}+\alpha_{c}}\right)}^{2}\right] & \mathrm{for} & \lambda ≧\lambda_{c}. \end{array}\right.$$

From fitting the monochromator measurements with Equation (Equation 4), the center wavelength $\lambda_{c}$, FWHM bandwidth $\gamma_{c}$, and the asymmetry parameter $\alpha_{c}$ for each channel are determined. The FWHM bandwidth is corrected for the additional broadening from the finite bandwidth of the monochromator, however, this correction is found to be smaller than the wavelength calibration uncertainty of the monochromator. The SRF parameters found from this spectral calibration are plotted for all the spectroradiometer channels in Figure 3. The center wavelength $\lambda_{c}$ for our spectroradiometer is plotted with respect to the previous calibration by the manufacturer. Because the previous calibration predated these monochromator measurements by several years, differences are observed in the range of −4–5 nm. Since these differences are of a similar magnitude as the spectral resolution of the spectroradiometer, it highlights the need for these SRF calibrations [20]. The uncertainty contribution of the spectral calibration to the radiometric calibration is presented in Section 7.

Additional features in the SRF measurement results can be observed, for example, two small peaks in the SWIR-2 detector array stand out at approximately 2000 nm and 2170 nm. These are due to stray light effects and have since been corrected by the manufacturer with an additional filter [21].

## 5. Temperature Sensitivity

Because only the SWIR detector arrays of the spectroradiometer are temperature-stabilized, the response of the VNIR detector array is sensitive to temperature variations. This is a common issue, which is also found in other spectroradiometers [3,22,23], due to the temperature dependence of the quantum efficiency of silicon photodiode arrays [24]. This effect is most notable at the start of a measurement series when the spectroradiometer has just been powered up and is warming up from its internal heating. In a laboratory environment, the VNIR detector array in our spectroradiometer requires approximately 2 h to reach a stable operating temperature within the 0.1 ${}^{{}^{\circ}}C$ resolution of the internal temperature sensor. Temperature changes during field measurements will also introduce systematic errors in the VNIR range, but these can be corrected when the temperature sensitivity coefficients $C_{T,c}$ are known.

The temperature sensitivity is measured by mounting the spectroradiometer in the temperature-controlled enclosure of Figure 1. The air inside this enclosure is maintained at a desired temperature between 5 ${}^{{}^{\circ}}C$ and 45 ${}^{{}^{\circ}}C$ with a set of fans and a radiator. A coolant is circulated through this radiator by a Huber CC-K6 cooling and heating bath, while a Pt100 temperature sensor provides the feedback to the temperature controller of the air temperature. The fiber optic from the spectroradiometer is directly connected to an integrating sphere outside of the enclosure that provides a broadband and homogeneous illumination. The operating temperature of the spectroradiometer is then stepwise increased and decreased. After each step, the spectroradiometer is allowed to thermally stabilize and repeated signals are recorded. The relative change in the signal of the VNIR channels shows a linear relation as a function of the VNIR detector temperature as illustrated in Figure 4. Note that the environmental air temperature is 7 °C to 9 °C lower than the temperature recorded by the VNIR temperature sensor due to internal heating in the spectroradiometer.

From the linear fits of the spectroradiometer signals, the temperature sensitivity coefficients are determined for a reference temperature of 32.3 °C. The temperature sensitivity coefficients $C_{T,c}$ are plotted in Figure 5 and show strong similarities to those reported in [23,25].

Another temperature dependence effect can be found in a spectral shift of the SRF, for instance due to a temperature sensitivity of the reflective gratings in the spectroradiometer. This is investigated with the same setup, but instead of a broadband illumination, a mercury spectral line lamp is installed in the integrating sphere. Shifts in the spectroradiometer response to the mercury-vapor emission lines are observed this way when varying the spectroradiometer’s environmental temperature from 10 °C to 40 °C. However, the spectral shifts in this temperature range are found to be below our spectral calibration uncertainty of 0.1 nm.

## 6. Non-Linearity

The instrument model of Equation (Equation 1) assumes that the spectroradiometer has a linear response. To estimate the measurement uncertainty from this approximation, the non-linearity of the spectroradiometer response over its dynamic range is investigated. When measuring a stable broadband source at different integration times, a linear response would mean that the scaled signal $S_{c}/t_{\mathrm{int}}$ for each channel is constant. The deviation from this constant behavior is investigated by normalizing these scaled signals with those at a reference integration time $S_{c,\mathrm{ref}}/t_{\mathrm{int},\mathrm{ref}}$. The results from such measurements with a stable QTH lamp source are plotted for all channels of the detector array in Figure 6.

The largest deviations from a linear response are caused by detector saturation, as seen in Figure 6 where the VNIR and SWIR-1 signals saturate at approximately 31,000 DN and 22,000 DN, respectively. Within the dynamic range of the detectors, the VNIR and SWIR-2 detector arrays show the most significant non-linearity. The linear response approximation thus introduces an additional uncertainty $u_{\mathrm{non}-\mathrm{lin},c}$, which is estimated here as the bounds of the relative non-linearity for each channel. This uncertainty contribution in radiance transfer measurements is plotted in Section 9.

## 7. Radiometric Calibration

The radiometric responsivity of the spectroradiometer is calibrated with the setup sketched in Figure 1. The calibrated radiance $L_{\mathrm{ref}}$ of the RASTA is interpolated to the spectroradiometer wavelengths $\lambda_{c}$. The spectroradiometer responsivity is then calculated from the mean signal using $R_{c}=\frac{S_{c}}{t_{\mathrm{int},d}L_{\mathrm{ref}}\left(\lambda_{c}\right)}$.

The radiometric responsivities $R_{c}$ are plotted in Figure 7 for the custom fiber-bundle optics in comparison to a 4°-FOV lens fore-optic. The same radiance standard setup is applied to calibrate the spectroradiometer with the 4°-FOV lens fore-optic, albeit without the temperature-controlled enclosure. The fiber optics bundle with an OAP mirror reduces the overall responsivity to about half compared to the lens fore-optic on the spectroradiometer due to a decreased transmission. However, the reduced signal-to-noise ratio of this custom setup does not limit our radiance transfer measurement uncertainties, as will be shown in Section 9.

The relative combined standard uncertainty of the radiometric responsivity $u\left(R_{c}\right)/R_{c}$ is calculated according to [26], using the equation
(5)$${\left[\frac{u\left(R_{c}\right)}{R_{c}}\right]}^{2}={\left[\frac{u\left(S_{c}\right)}{S_{c}}\right]}^{2}+{\left[{\left.\frac{u\left(L_{\mathrm{ref}}\right)}{L_{\mathrm{ref}}}\right|}_{\lambda_{c}}\right]}^{2}+{\left[{\left|\frac{\nabla L_{\mathrm{ref}}}{L_{\mathrm{ref}}}\right|}_{\lambda_{c}}u\left(\lambda_{c}\right)\right]}^{2}.$$

The first term in the above equation is from the standard uncertainty of the signal mean $u\left(S_{c}\right)$, i.e., the measurement noise. The second term is the relative uncertainty of RASTA interpolated to the center wavelengths $\lambda_{c}$. The last term comes from the propagated uncertainty in the center wavelength $u(\lambda_{c})$ from the spectral calibrations, discussed in Section 4, and depends on the spectral gradient of the radiance standard $\nabla L_{\mathrm{ref}}$. These three uncertainty contributions are plotted in Figure 8. The contributions from the standard uncertainty of the mean $u\left(S_{c}\right)$ of 50 repeated measurements and the uncertainty contribution from the spectral calibration are both below the relative uncertainty of RASTA.

## 8. Polarization Sensitivity

The response of the spectroradiometer depends on the polarization of the incoming radiation. This polarization sensitivity is caused for the most part by the gratings and beam splitters that separate the light paths for the three detector arrays. While the light from integrating spheres illuminated by QTH lamps can be considered unpolarized, partially polarized light can have an impact on field measurements [27].

The polarization sensitivity of the HR-1024i spectroradiometer is characterized with the 4°-FOV lens fore-optic installed. The response to linear-polarized light is measured with a rotating broadband polarizer (LOT-QuantumDesign, UBB01A) in front of the aperture of a broadband integrating sphere. The spectroradiometer response shows a dependence on the polarization angle with respect to the slit orientation. Figure 9 illustrates this with plots of the signals from selected channels from the three detector arrays.

The polarization sensitivity is determined by fitting the spectroradiometer signal as a function of the polarization angle ϕ. Following Malus’ law, the signal can be described via [13]:(6)$$S_{c}=S_{c,o}+2A_{c}\left[\cos{\left(\phi -\phi_{c,0}\right)}^{2}-\frac{1}{2}\right],$$
where $S_{c,o}$ is the signal mean, $A_{c}$ is the polarization-dependent response, and $\phi_{c,0}$ is the polarization angle at which the spectroradiometer has its maximum signal. The relative polarization sensitivity of the spectroradiometer is then quantified as $P_{c}=\frac{A_{c}}{S_{c,o}}$. The results of fitting each channel with Equation (Equation 6) are plotted in Figure 10. While polarization sensitivity is observed over the entire wavelength range of the spectroradiometer, the largest sensitivities are at the wavelengths where the channels between detector arrays overlap.

## 9. Radiance Transfer

The calibrated spectroradiometer can be applied to measure the radiance from an homogeneous source using the parameters determined in the previous measurements. An example of such a radiance transfer measurement to a large-aperture integrating sphere is shown in Figure 11. The integrating sphere is here the source for radiometric calibrations of a device under test (DUT), where the spectral radiance from the integrating sphere aperture is determined from the simultaneous measurements with the spectroradiometer. The FOV of the spectroradiometer is aligned with the same method described in Section 2 to measure the radiance from the integrating sphere wall that is opposite the center of the rectangular aperture.

The integrating sphere radiance $L(\lambda_{c})$ is determined from the measured spectroradiometer signal $S_{c}$ via
(7)$$L\left(\lambda_{c}\right)=\frac{S_{c}}{R_{c}t_{\mathrm{int},d}\left[1+C_{T,c}\left(T_{d}-T_{\mathrm{ref}}\right)\right]}.$$

The combined standard uncertainty in the radiance measurement is calculated with [26]
(8)$${\left[\frac{u\left(L(\lambda_{c})\right)}{L(\lambda_{c})}\right]}^{2}={\left[\frac{u\left(S_{c}\right)}{S_{c}}\right]}^{2}+{\left[\frac{u\left(R_{c}\right)}{R_{c}}\right]}^{2}+{\left[\frac{u\left(C_{T,c}\right)\left(T_{d}-T_{\mathrm{ref}}\right)}{1+C_{T,c}\left(T_{d}-T_{\mathrm{ref}}\right)}\right]}^{2}+\frac{u_{\mathrm{non}-\mathrm{lin},c}^{2}}{3},$$

Each of the uncertainty contributions of Equation (Equation 8) is also plotted in Figure 12. Again, the radiometric calibration uncertainty is the dominant contribution for the majority of the spectroradiometer channels. The non-linearity uncertainty $u_{\mathrm{non}-\mathrm{lin},c}$ is only for the VNIR detector array significantly larger than the measurement noise and exceeds the radiometric responsivity uncertainty at wavelengths below 380 nm and around 1000 nm. For the wavelength range from 380 nm to 2500 nm, the relative expanded uncertainty (k=2) of this radiance measurement is between 1% and 3%.

The cross-over region between the VNIR and SWIR-1 detector arrays at approximately 1000 nm shows differences in the radiance measurements that are larger than the expanded uncertainty, as also seen in the radiance measurements in Figure 12. This indicates that Equation (Equation 7) does not accurately model the response at the edges of the detector arrays. One of the causes could be found in the linear response approximation. Non-linearity correction functions could be determined for each channel, similar to the methods described in [28,29], in order to account for the observed discontinuity between the VNIR and SWIR-1 detector arrays. In the cross-over region of the SWIR-1 and SWIR-2 detector arrays on the other hand, the radiance measurements overlap at approximately 1900 nm.

## 10. Discussion and Conclusions

In the presented radiance transfer measurements, the combined standard uncertainties are dominated across the majority of the wavelength range by the uncertainties of the DLR radiance standard. The spectroradiometer can thus be applied as a radiance transfer instrument when following the discussed calibrations without a significant impact on the total uncertainty budget for the wavelengths between 380 nm and 2500 nm. Uncertainties above 1% are found at the edges of the VNIR detector array, i.e., below 380 nm and at approximately 1000 nm, as well as above 1650 nm due to the higher uncertainties in RASTA.

Because other spectroradiometer models are based on similar optical designs and detector arrays, the calibration results presented here resemble reports on other spectroradiometers [2,3,4]. However, the geometric characterization has not been discussed in this work since all the measurements here apply spatially homogeneous light sources that overfill the spectroradiometer aperture. In order to extend the presented methods and results to non-uniform sources, as are commonly encountered in field measurements, the field-of-view of the spectroradiometer channels will have to be taken into account [30].

Spectral straylight effects have also not yet been investigated for this spectroradiometer, but could be characterized with a set of filters with known spectral transmissions [31]. Because the QTH lamps involved in the radiance transfer measurements have almost identical filament temperatures, any straylight corrections are only expected to be significant in the wavelength region <300 nm [32].

The main highlight from this work is that relative uncertainties at a level of 0.1% in radiance transfer measurements could be readily achieved with our spectroradiometer when radiance standard uncertainties are reduced by an order of magnitude.

## Figures and Tables

**Figure 1 sensors-23-02339-f001:**
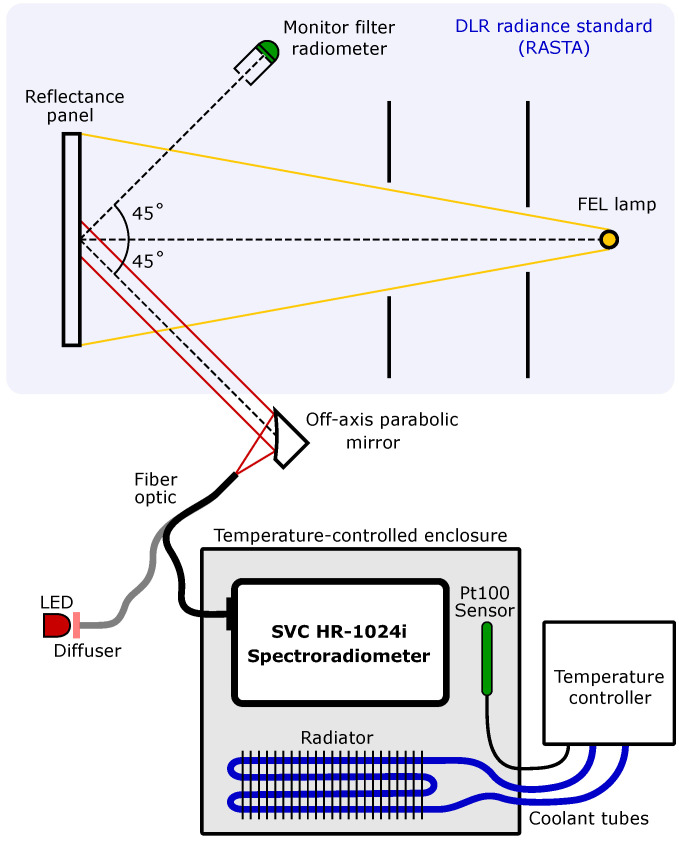
Radiometric response calibration setup of the spectroradiometer on the DLR radiance standard (RASTA). The calibrated radiance from a reflectance panel illuminated by an FEL lamp is monitored by five filter radiometers, but only one radiometer is shown in the sketch for clarity. The spectroradiometer is calibrated inside a temperature-controlled enclosure where the air temperature is maintained by a radiator and monitored by a Pt100 temperature sensor connected to a temperature-controller. The fiber optic bundle from the spectroradiometer is attached to an off-axis parabolic mirror that reduces the spectroradiometer’s field of view to approximately 4${}^{{}^{\circ}}$. The spectroradiometer’s field of view is aligned to the center of the RASTA reflectance panel with the off-axis parabolic mirror. This is aligned before the radiometric calibration by coupling the light from a diffused LED source into the fiber optic instead of the spectroradiometer.

**Figure 2 sensors-23-02339-f002:**
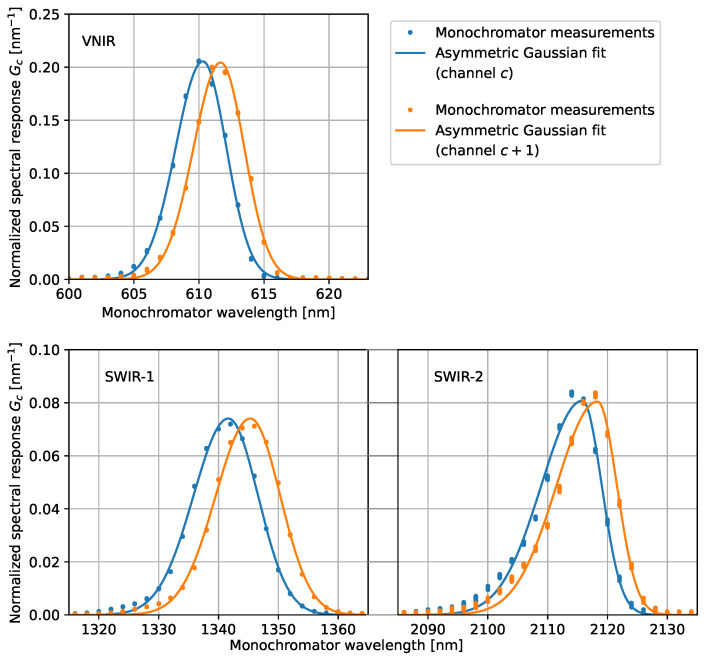
Normalized spectral response functions of two adjacent channels in each of the spectroradiometer detector arrays with asymmetric Gaussian fits. The plotted channel *c* is number 190, 610 and 850 for the VNIR, SWIR-1, and SWIR-2 detector, respectively. The asymmetry of the SRF is most pronounced in the SWIR-2 channels.

**Figure 3 sensors-23-02339-f003:**
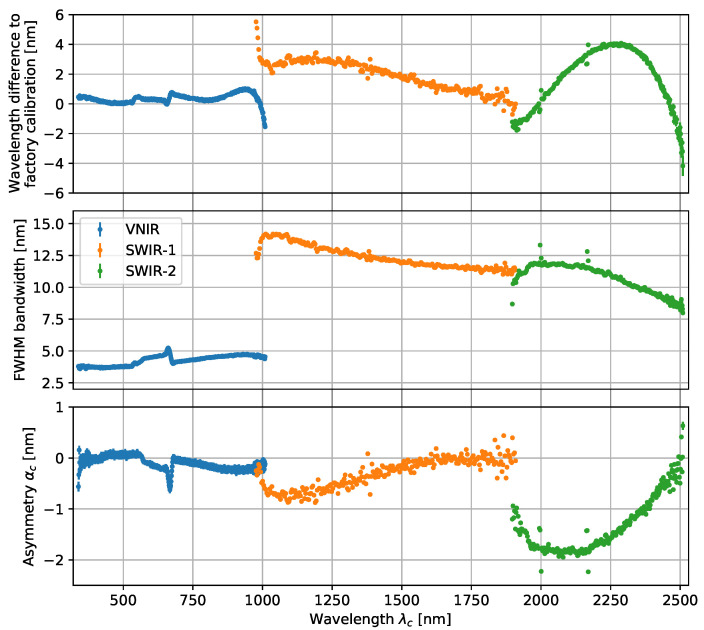
Spectral calibration results from an asymmetric Gaussian spectral response fits for all spectroradiometer channels. (**top**) Difference between the center wavelength $\lambda_{c}$ and the factory calibration from several years prior; (**middle**) full width at half maximum of the spectral response function; and (**bottom**) asymmetry parameter from the spectral response fits.

**Figure 4 sensors-23-02339-f004:**
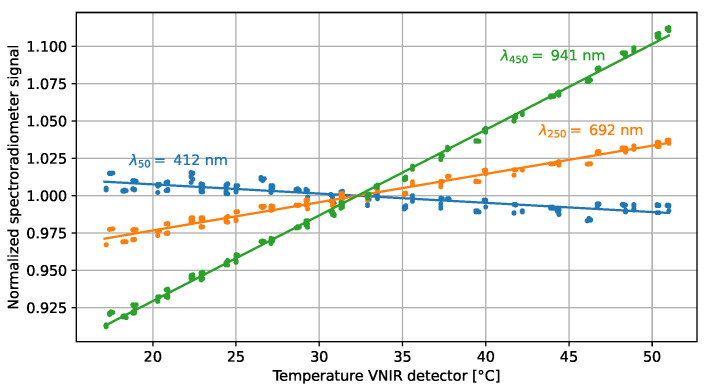
Temperature sensitivity measurements and linear fits of three channels in the VNIR detector array of the spectroradiometer. The spectroradiometer signals have been normalized to the response at a reference detector temperature of 32.3 °C. The labels indicate the center wavelength $\lambda_{c}$ of the plotted channel.

**Figure 5 sensors-23-02339-f005:**
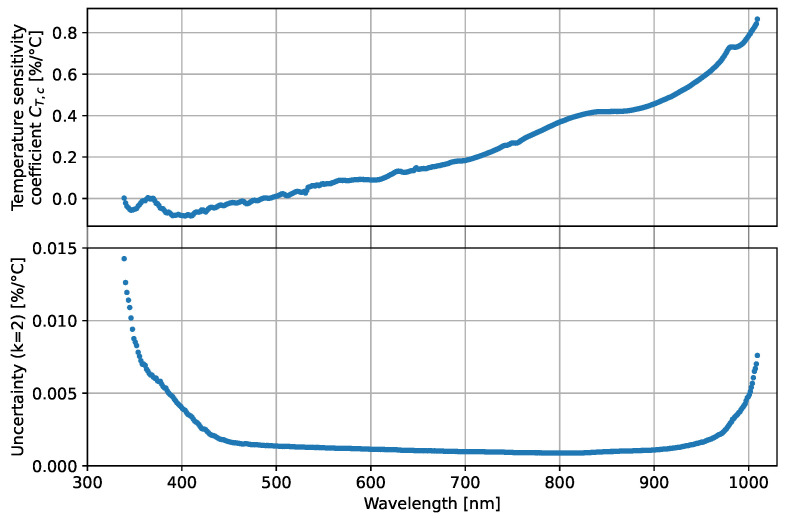
Temperature sensitivity coefficients and expanded uncertainties of the spectroradiometer’s VNIR channels.

**Figure 6 sensors-23-02339-f006:**
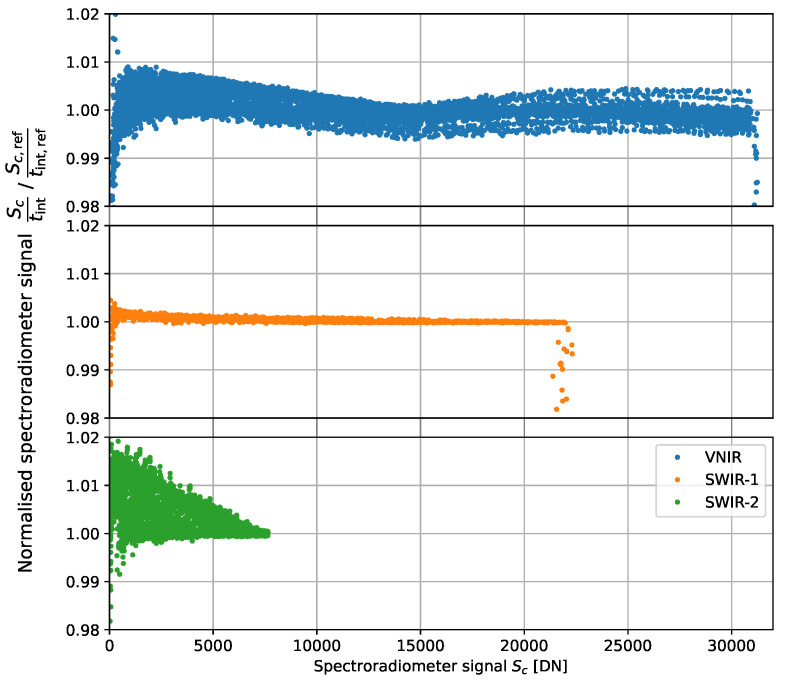
Spectroradiometer signals $S_{c}$ of each detector array scaled by its integration time $t_{\mathrm{int}}$ and normalized to the reference signal $S_{c,\mathrm{ref}}$ at reference integration times $t_{\mathrm{int},\mathrm{ref}}$ = 80 ms, 40 ms, and 15 ms for the VNIR, SWIR-1, and SWIR-2 detectors, respectively.

**Figure 7 sensors-23-02339-f007:**
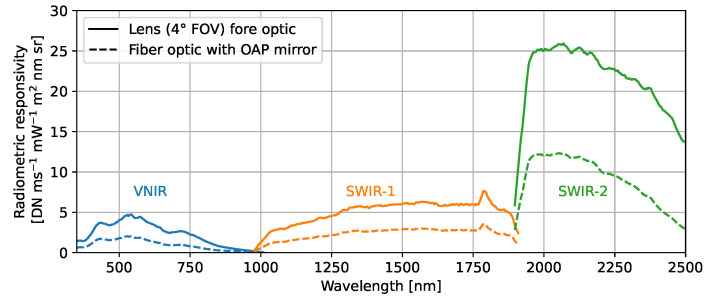
Radiometric responsivity of the spectroradiometer from calibrations with either a 4${}^{{}^{\circ}}$ FOV lens or the fiber bundle with an off-axis parabolic (OAP) mirror as applied in the radiance transfer measurements.

**Figure 8 sensors-23-02339-f008:**
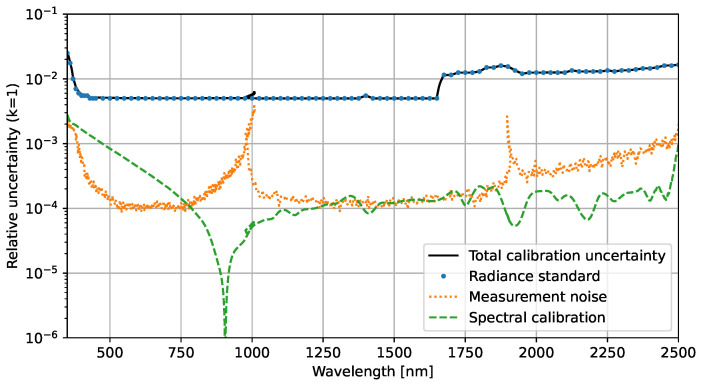
Contributions to the relative uncertainty in the radiometric calibration of the spectroradiometer. The DLR radiance standard (RASTA) has the most significant uncertainty over the spectroradiometer’s wavelength range.

**Figure 9 sensors-23-02339-f009:**
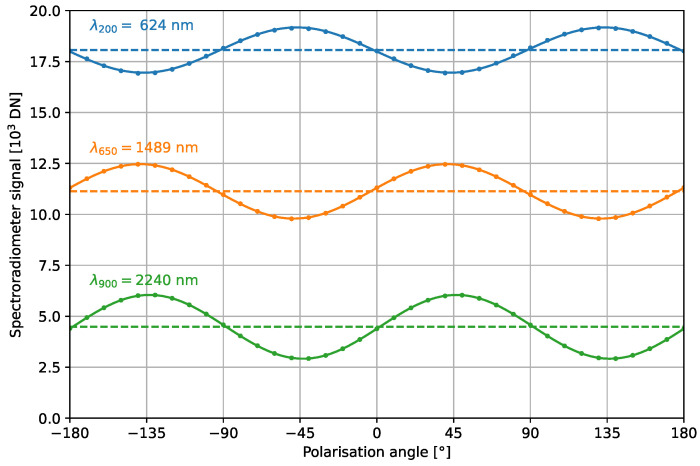
Spectroradiometer response of three channels to linear-polarized light as a function of the polarization angle with respect to the spectroradiometer slit orientation. The polarization sensitive response follows Malus’s law, as shown by the fits with Equation (Equation 6).

**Figure 10 sensors-23-02339-f010:**
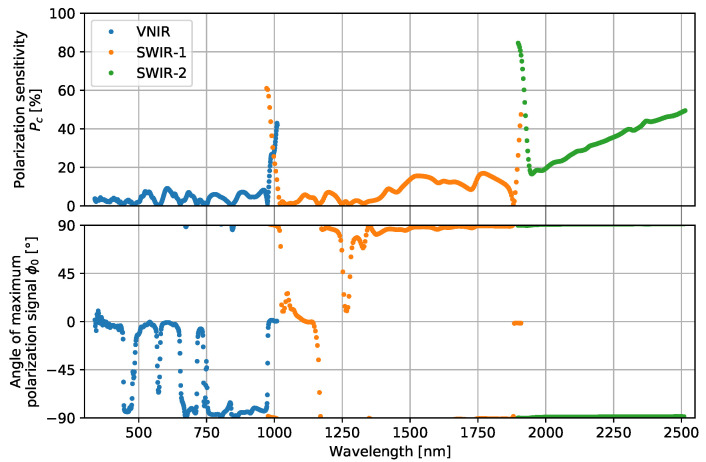
(**top**) Polarization sensitivity of the SVC HR-1024i spectroradiometer. (**bottom**) Angle of polarization with respect to the slit direction of the spectroradiometer where its signal is maximum.

**Figure 11 sensors-23-02339-f011:**
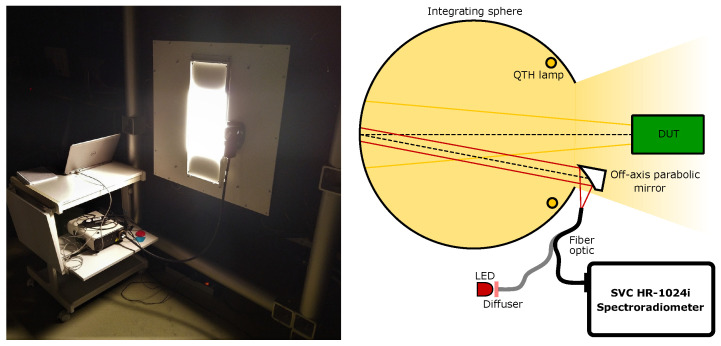
Radiance transfer setup where the calibrated spectroradiometer measures the spectral radiance of a large-aperture integrating sphere. (**left**) Photograph of the integrating sphere aperture with the spectroradiometer and the fiber-optic bundle. (**right**) The field-of-view (FOV) of the spectroradiometer is aligned to the center of the back of the integrating sphere, such that it overlaps with the FOV of a device under test (DUT).

**Figure 12 sensors-23-02339-f012:**
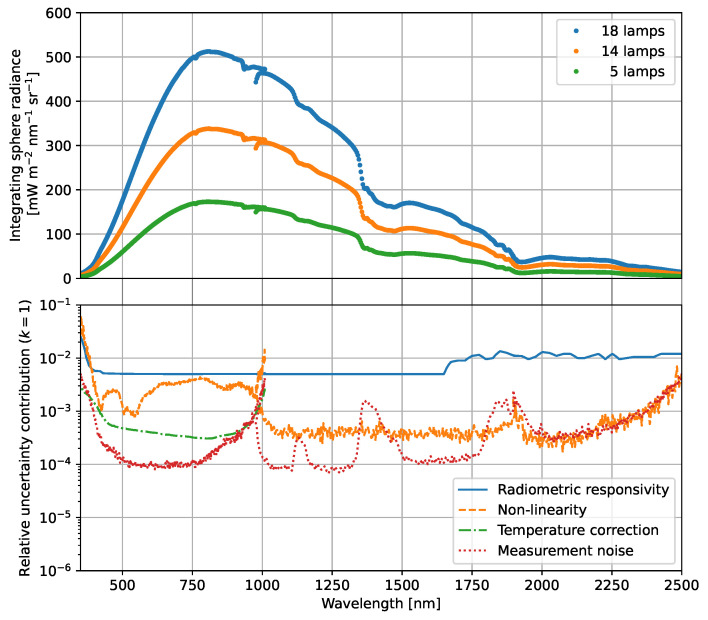
(**top**) Radiance from the integrating sphere at different lamp combinations as measured with the calibrated spectroradiometer. (**bottom**) Uncertainty contributions in the spectroradiometer measurements.

**Table 1 sensors-23-02339-t001:** Detector arrays of the SVC HR-1024i spectroradiometer applied for radiance transfer measurements.

Detector (Type)	Wavelength Range (nm)	Number of Channels	Temperature Stabilization
VNIR (Si)	339–1009	512	none
SWIR-1 (InGaAs)	977–1908	256	−5 ${}^{{}^{\circ}}C$
SWIR-2 (extended InGaAs)	1896–2510	256	−10 ${}^{{}^{\circ}}C$

## Data Availability

The data presented in this study are available on request from the corresponding author.

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
