# Peer review of "Spectroradiometer Calibration for Radiance Transfer Measurements"

_sensors, 2023, doi:10.3390/s23042339_

Round 1
Reviewer 1 Report
Very well written description.
Some additional information about the set of five filter radiometers could be provided (even if available in other publications).
Check diameter in line #54.
Check missing unit in line #66.
Author Response
The authors thank the reviewer for the suggestions and have made the following changes to the manuscript (line numbers refer to the updated version):
- The description of the filter radiometers is expanded in lines #59-61.
- To clarify the beam diameter the sentence in lines #54-55 is rephrased.
- Fixed the issue of missing units in line #68.

Reviewer 2 Report
The manuscript describes the calibration behavior of the specific spectroradiometer and how the results can be applied to make radiance transfer measurements. In my opinion the text is well structured and easy to follow. I also believe that the results are interesting to people working with the instrument calibration work. I just have a couple of minor comments:
L21-22: "On the one hand", could be left out
Fig. 2: The adjacent channels are most likely marked with blue and orange colors. The legend should also reflect this.
Fig. 4: Do the colors also reflect the different sensors? this should be stated in the legend.
Author Response
The authors thank the reviewer for the comments and made the following changes to the manuscript:
- Removed "On the one hand" in line #21-22 as suggested.
- The legend and caption of Fig.2 are updated to clarify the differences between the plot colors.
- The colors in Fig.4 correspond to different channels of the same detector. A sentence has been added to the figure caption to better clarify this.

Reviewer 3 Report
This is a very well written manuscript that details the calibration of an SVC spectroradiometer. It provides the sound scientific basis for a way to reduce the uncertainty in the spectroradiometer measurements to ~1% with the limitation in the applied radiance standard. A reasonable further reduction of uncertainty to ~0.1% is demonstrated if radiance standard uncertainties are reduced by a factor of 10.
The only significant (but minor) flaw is in the manuscript's structure. It has an introduction section, then seven calibration procedure and parameter sections, then an application section, followed by a short "discussion" section that is really just a summary. There is no "conclusion" section to highlight the somewhat novel finding of "a reasonable further reduction of uncertainty to ~0.1% is demonstrated if radiance standard uncertainties are reduced by a factor of 10." I suggest the final section (10) be renamed to "Discussion and Conclusion," and the final sentence in lines 251 to 254 be highlight as the concluding finding of interest.
Other very minor tidbits:
Table 1: the wavelength range should have the units labeled as nm.
line 221: what are the units of the expanded uncertainty 1;3? Percent?
line 236: 380;2500 should be followed by nm.
Author Response
The authors agree with the reviewer's assessment of the highlighted finding. Because the reduction in uncertainty is not directly demonstrated, it is stated as a future direction of the work. As suggested by the reviewer, we renamed the last section to "Discussion and Conclusion" and rephrased the section to make the final sentence stand out more.
The authors also fixed the issue of missing units in Table 1, line #222 and line #237 (line numbers refer to the updated manuscript).
